# Exercise Prescription to Foster Health and Well-Being: A Behavioral Approach to Transform Barriers into Opportunities

**DOI:** 10.3390/ijerph18030968

**Published:** 2021-01-22

**Authors:** Daniela Lucini, Massimo Pagani

**Affiliations:** 1BIOMETRA Department, University of Milan, 20122 Milano, Italy; massimo.paganiz@gmail.com; 2Exercise Medicine Unit, Humanitas Clinical and Research Center, 20089 Rozzano, Italy

**Keywords:** exercise prescription, stress, wellbeing, behavioral medicine, physical activity, functional syndromes

## Abstract

The current literature contains multiple examples of exercise interventions to foster health and to prevent/treat many chronic non-communicable diseases; stress and functional syndromes. On the other hand, sedentariness is increasing and to transform a sedentary subject into a regular exerciser is not only very difficult but considered by some unrealistic in current clinical practice. Ideally a physical activity intervention may be considered actually efficacious when it outgrows the research setting and becomes embedded in a system, ensuring maintenance and sustainability of its health benefits. Physicians need specific skills to improve patients’ exercise habits. These range from traditional clinical competencies, to technical competencies to correctly prescribe exercise, to competencies in behavioral medicine to motivate the subject. From a behavioral and medical point of view, an exercise prescription may be considered correct only if the subject actually performs the prescribed exercise and this results in an improvement of physiological mechanisms such as endocrine, immunological and autonomic controls. Here we describe a model of intervention intended to nurture exercise prescription in everyday clinical setting. It aims to a tailored prescription, starts from the subject’s assessment, continues defining clinical goals/possible limitations and ends when the subject is performing exercise obtaining results.

## 1. Introduction

Prevention and treatment of chronic non-communicable diseases (CNCD) represent one of the major goals and challenges of governments and medical institutions worldwide. Interventions aiming to improve lifestyles play a pivotal role in this scenario since ancient times when Hippocrates (5th century BC) wrote: “*if we could give every individual the right amount of nourishment and exercise, not too little and not too much, we would have found the safest way to health*” [1]. This statement underlies not only the importance of nutrition and exercise, but almost the importance to titrate their “dose”, as if they were a medicine. Eons of interest and scientific research eventually followed through, and the role of regular physical activity in influencing health grew of importance so much as to be proffered as a possible therapy [2] at the end of nineties, being subsequently considered itself an official treatment ten years later when the scientific community launched the slogan “exercise is medicine” [3], and being considered today a sustainable tool both for individual health and community [4] and a real tool to foster health and well-being in the general population and not only in patients affected by chronic diseases [5].

The importance to address lifestyle in the prevention and management of CNCD is also corroborated by the observation that social environments strongly influence individual behavioral choices henceforth affecting health [6]. The link between social ties and behavior is well-known and it may be crucial both in worsening health (for instance, network phenomena appear to be relevant to the biologic and behavioral trait of obesity, and obesity appears to spread through social ties [7]) and in improving it (for instance sport participation improves overall health [8]). On the other hand, the benefits of a healthy lifestyle, and in particular of exercise, are not limited to the prevention/management of chronic diseases but are also related to the improvement of social relationships, socialization, reduction of illegal behaviors, reduction of isolation and depression, stress management and improvement of work and scholastic performances [9,10,11,12,13].These considerations immediately suggest that successful interventions reduce incidence of chronic diseases and in general foster health. Complex approaches are necessary to reach this goal; they need to consider actions both at individual and social level, involving different professional skills, different community settings where people live, and different methodologies considering also the utilization of new technologies [14,15,16,17,18].

The individual benefits of regular exercise are evident both in secondary, in primary and primordial prevention [16,17,18,19,20,21,22,23,24,25,26,27,28] of many diseases, from cardiometabolic conditions (coronary artery diseases, hypertension, heart failure, diabetes, etc) to cancer, functional diseases as chronic fatigue syndrome, fibromyalgia, and many other illnesses. Furthermore regular exercise is associated with longevity, happiness, reduced risk of physical disability and dependence and in general with reduction in all-cause mortality [10,29,30,31,32,33]. Of particular interest is the consideration that cardiorespiratory fitness may nowadays be considered an important quantitative predictor of all-cause mortality and it is a potentially stronger predictor of mortality than established risk factors [30]. From a clinical point of view, to help a subject to improve her/his cardiorespiratory fitness starting from a low level (corresponding to the capacity to perform an exercise lower than six METs) to a moderate level (corresponding to the capacity to perform an exercise of 6–8 METs) means to help her/him to get the largest observed reduction in mortality [34,35]. A positive relationship between lean body mass and longevity is also described, particularly in patients with low body mass index [32]; moreover a low muscle strength was more strongly and significantly associated with all-cause mortality than low muscle mass [31].

The mechanisms underlying the relationship between health and regular physical exercise (and in general a healthy lifestyle) are numerous and complex. Exercise may act on major regulatory systems controlling immunological functions, hormones and autonomic nervous system regulation [24,36,37,38,39,40,41]. It may produce benefits similar to drugs in secondary prevention of coronary heart disease, rehabilitation after stroke, treatment of heart failure, and prevention of diabetes. New findings suggesting new benefits are constantly emerging, like the observation that cardiorespiratory fitness is also a predictor of intestinal microbial diversity, suggesting the use of exercise prescription as an adjuvant therapy in combating dysbiosis-associated diseases [42]. Of particular interest are the beneficial effects of aerobic exercise on cardiac autonomic control [36,37] and the possibility to assess it employing non-invasive and sustainable methodologies such as spectral analysis of heart rate variability [4,43,44]. Moreover, the positive effects of exercise and in general of healthy lifestyle might counterbalance a high genetic coronary risk [45]. The relationship between genetics and lifestyle is particularly intriguing. In fact, exercise, healthy nutrition and even stress reduction could produce possible regulatory effects on gene expression [46]. Adherence to a low-risk lifestyle could prolong life expectancy [29]. Interventions based on exercise and healthy nutrition are capable even to promote remissions of type 2 diabetes in obese patients [22] and to improve perioperative outcomes and to reduce post-surgical complications [47].

Of fundamental value is also the observation that exercise may help to reduce other cardiometabolic risk factors like overweight, smoke, stress, altered lipid profile, etc. [2,3,9,20,48,49] and its effects in prevention/treatment of diseases are particularly efficacious when associated to other healthy lifestyle elements (especially healthy nutrition, stop smoking, stress management) [16,18,19,45].

Exercise and physical activity (see Table 1) are not the mere opposite of sedentariness. In fact many studies show that sedentary behavior is per sè an important cardiometabolic risk factor [28,50,51] and that physical activity may attenuate the detrimental effects of sedentary behavior. Conversely sedentary behavior may reduce the beneficial effects of physical activity. Then for a clinical perspective, interventions need to be aimed both to increase time dedicated to exercise and to reduce time dedicated to sedentary activities as sitting time [50].

## 2. Sport and Exercise Medicine

Sport and exercise are sometimes used to indicate the same concept, risking confusion. In fact these terms have different meanings (see Table 1). From a clinical point of view their main difference may perhaps be summarized with the following sentence: the goal of sport is competition while the goal of exercise is health. Consequently the goal of sports medicine may be considered to help the athlete to prevent and to manage injuries, to maintain health avoiding any problem, and to increase performance. The goal of exercise medicine may be considered, instead, to utilize physical exercise as a tool to foster health and wellbeing, to prevent and manage most chronic non communicable diseases in any subject independently of her/his fitness level. Sport medicine physicians generally work with athletes used to the ontology, benefits/problems and practical issues of exercise, willing (and usually fit) to perform exercise even at high intensity. Conversely Exercise medicine physicians usually approach subjects who actually do not exercise, and who need an exercise prescription tailored on their own characteristics, needs and clinical goals [24]. The beneficial effects of exercise may be increased by addressing also other components of lifestyle, in particular nutrition, stop smoking and stress management. This approach is typical of lifestyle medicine [61], which consists in the therapeutic use of evidence-based lifestyle interventions to treat and prevent lifestyle related diseases in a clinical setting, empowering individuals with the knowledge/skills to make effective behavior changes.

There is a great debate [62,63,64] on who is the professional entitled to practice exercise medicine (in particular exercise prescription) [62,63,64,65] and this debate becomes even more heated when considering the different education training of professionals who may have a role in exercise medicine in the different countries. Sport physicians, exercise physicians, internists, cardiologists, rehabilitation physicians, endocrinologists, general practitioners, etc, among medical physicians, may have a role. Moreover other exercise professionals, like exercise physiologists [63], and physiotherapists [66] can claim to be entitled to a significant role in the field of exercise medicine. One possible approach to overcome this debate, may be to consider a real collaboration between different professionals with a common interest on prevention and exercise, sharing significant learning, then creating a new knowledge, as speculated in the paper by Gates [62,65].

It is important to evidence the specific competences that are required in order tprescribe exercise or more in general a life style change program [4,62,67,68,69]. They may be summarized as follows:Conventional medical expertise is needed to adequately assess subjects/patients, to know medical implications of the disease to prevent/treat and of its eventual pharmacological treatment, to tailor exercise and nutrition prescriptions to subjects’/patients’ needs and characteristics, to reveal possible contraindications for some modalities/intensities of exercise.Psychological expertise is needed to motivate and help patients to change behavior and/or to manage possible psychological issues resulting from a major disease (if present) [70].Technical expertise is needed in order to prescribe exercise, nutrition or for smoking cessation programs, to educate subjects/patients and to help them overcome practical barriers.

To acquire all these competencies may be more important than simply belonging to a specific profession or medical specialization. To include lifestyle medicine in undergraduate medical curricula [67,68,69,71] or in other health professionals’ curricula, would be welcomed in order to implement efficacious lifestyle change programs. This improvement in medical training would help to overcome an important limitation often encountered in clinical practice: the limitation of the physician’s role to subject/patient assessment and or to pharmacological/surgical management of the disease. On the contrary physician’s role should include the ability to prescribe exercise and to reveal clinical issues (for instance the presence of contraindications to some exercise modality or the need to consider the effects of some pharmacological treatment) mandatory to an efficacious and safe tailored, inclusive, prescription of exercise programs.

Other exercise professionals (exercise physiologists, physiotherapists) have an important specific role, especially the one of following up the subject/patient over time, adjusting the program, designing new protocols, etc. This approach may contribute to eradicate interprofessional barriers constructed as a result of historical professional hierarchies and tensions [65], hence opening a new opportunity to implement exercise medicine [65].

## 3. Models to Implement Exercise into Lifestyle Programs

Despite the benefits of physical activity are well documented, physical inactivity is rapidly becoming a major global concern and now this issue is described as pandemic, with health, economic, environmental and social consequences [50,51,72,73]. As previously reported, any lifestyle intervention considering physical activity must consider two parallel lines of intervention. The first one aims to reduce sedentariness (to take advantage of any opportunity during day life to perform physical activity, such as avoiding elevators, preferring stairs, etc), and the second one aims to introduce into daily life a structured exercise program [28].

The scientific literature contains many examples of physical activity interventions: some of which are efficacious and may represent a model to scale up [14,74,75]. Other ones resulted uncapable to produceany significant change [14,15,76,77]. The programs that appeared to be most effective, are characterized by multifactorial interventions and are designed for individuals or groups according to specific characteristics and needs [14,16,17,78,79], often including computer/technology-based strategies [14,16]. Simple counseling about healthy behavior, in particular exercise, yields limited or no results [14,15,76,77], while success requires the implementation of specific action plans in partnership with subjects/patients and intentional follow-up [4,67,74,75]. Ideally a physical activity intervention may be considered actually efficacious when it outgrows the research setting and becomes embedded in a delivery system, ensuring maintenance and sustainability of its health benefits [4,75].

Approaches based on the combination of careful patient assessment, tailored prescription of healthy nutrition, exercise and smoking cessation plans, cognitive behavioral strategies (CBS), have been shown to be most effective [4,14,17,18]. These approaches (see also our previous paper [4]) require multidisciplinary expertise to be included in the program. This is often guaranteed by the involvement of several different healthcare professionals (specialists, general practitioners, psychologists, dieticians, exercise physiologists) which are required to work in truly collaborative and goal oriented teams in order to get positive results [14,18,74,75]. On the contrary, simple referral to other healthcare professionals without joint responsibility often results in a time consuming, expensive and rather ineffective approach [14,15,76]. Notably this strategy is frequently presented in clinical contexts outside research programs, representing paradoxically a barrier to a successful result. Other important barriers reported by patients regarding the financial burden and time investment [80,81], i.e., conditions woefully characterizing intensive lifestyle modification programs (multiple encounters with different experts, educational sessions, assisted training sessions, etc, that are not always covered by social security system or insurance). These elements should be minimized as much as possible in order to design realistic and efficacious programs. The possibility to have a physician trained in Life Style Medicine, as hypothesized in many papers [67,68,69,71], capable of entirely manage the patient’s pathway forward a new behavior, may help to overcome some of these barriers [4]. Obviously the possibility to add other different healthcare professionals will improve the quality of the program when economic and organizational barriers are not present.

## 4. Subject’s Empowerment and Behavioral Medicine as Tools to Prescribe Exercise

### 4.1. Empowerment

WHO defines empowerment as “a process through which people gain greater control over decisions and actions affecting their health and should be seen as both an individual and a community process. Empowerment in health care generally refers to the process that allows an individual or a community to gain the knowledge, skills, and attitude needed to make choices about their care” [82]. The key components of empowerment are: (1) understanding by the patient of his/her role; (2) acquisition by patients of sufficient knowledge to be able to engage with their healthcare provider; (3) patient skills and (s4) the presence of a facilitating environment. Translating these concepts into exercise medicine means to employ behavioral medicine tools (see Table 2) in order to help the subject/patient to understand the important link between his/her health issue and exercise, showing the real benefits associated to become physically active following prescription tailored on his/her needs, characteristics and preferences, hence defining a new ecology.

Behavioral medicine may be defined as “the interdisciplinary field concerned with the development and integration of behavioral, psychosocial, and biomedical science knowledge and techniques relevant to the understanding of health and illness, and the application of this knowledge and these techniques to prevention, diagnosis, treatment and rehabilitation” [83].

### 4.2. Multiplicity of the Psychological Theories and Models which Can Be Useful (See Table 3)

Motivational interviewing is an example of a methodology using a combination of behavior change techniques to help pragmatically people to change behavior [84]. It is a directive, client-centered method for enhancing intrinsic motivation by exploring and resolving ambivalence and barriers to behavior change considering lecturing or confrontation as unhelpful [85]. The main principles of motivational interviewing are: express empathy (through reflective listening); develop discrepancy (between the individual’s goals and their current behavior), avoid argumentation and roll with resistance (acknowledge and explore the individual’s resistance to change, rather than opposing it) [86,87].

Another approach commonly known as ‘nudging’, primarily drawn from behavioral economics, has attracted interest in these years. It aspires to ‘nudge’ people’s choices, not by taking off the less healthy ones, but by making the healthier option easier [88].

Effectively motivating patients to change behavior can be a frustrating and difficult challenge. Merely encouraging patients at the end of an office visit to attempt such changes yields limited results. Success requires the development of specific healthy lifestyle action plans in partnership with patients and intentional follow-up in subsequent visits [67]. Motivation may be considered as a strategy to help the subject/patient to transform his/her desires about health improvement into realistic goals, furnishing all the required resources in order to be proactive.

## 5. From Subject’s Assessment to Exercise Prescription

Tailored prescription of exercise programs is strongly recommended in all exercise guidelines [3,20,48] and requires the clear definition of modality, intensity, frequency, duration and progression of exercise in order to reach the set clinical goals considering also individual characteristics, limitations and preferences. A correct tailored exercise prescription requires a specific focus on the single patient/subject more than on a specific disease/condition. Obviously some specific clinical conditions require, sometimes, particular protocols; in addition, a correct tailored exercise prescription needs to consider, for a single patient, all the coexisting conditions and possible contraindications. The process which drives to a tailored prescription starts from the subject’s assessment, continues defining clinical goals and possible limitation (Figure 1) and ends when the subject is performing exercise.

### 5.1. Subject Assessment

A correct assessment of subjects’ clinical conditions and lifestyles is of paramount importance in order to define clinical goals and to exclude any contraindication to some exercise modality/intensity or possible development of complications, moreover it may furnish important parameters, for instance maximal and basal heart rate, necessary to a correct exercise prescription.

#### 5.1.1. Clinical Assessment

The need for formal medical screening (in order to verify the clearance to perform exercise and minimize possible health related problems) depends on the age, clinical condition, cardiac risk factor and the intensity of exercise [20,48,93]. Healthy adults who desire to perform low-moderate intensity walking program or equivalent exercise do not need a formal medical screening when the intention is to exclude possible cardiovascular events [20,48,93], but it is welcomed particularly when the intention is to tailor exercise prescription and to set specific clinical goal/s in order to reach specific benefits. In this case it may be useful to consider personal and parenteral history, standard medical examination (anthropometric and hemodynamic data), medical tests (for instance a cardiopulmonary stress test or a simple stress test in order to define training heart rate [20,30,48,93] depending on patients’ conditions, risk and intensity of physical activity). Moreover subjects who desire to perform a resistance training program should be carefully screened for both cardiovascular limitations and preexisting orthopedic and musculoskeletal problems [93].

#### 5.1.2. Lifestyle Assessment

Ad hoc questionnaire may be used in order to define lifestyle, with particular focus on physical activity, nutrition, perception of stress, substance abuse, alcohol consumption, smoke and sleep [94]. Specifically regarding the assessment of daily physical activity, it is particularly important to define for every modality of performed physical activity the intensity an duration which may permit (referring to validated tables [95]) to calculate (with reasonable accuracy) the total activity dose expressed in Metabolic equivalents (METs) of aerobic physical activity performed in a given period (one day or one week). Physical activity may also be quantified using objective methods like wearable devices (pedometer, accelerometers [94]). These techniques may be useful also to monitor improvement of physical activity level or to help motivate adherence to the prescribed program. Correct subject’s assessment is also an essential starting point of Cognitive Behavioral Strategies (CBS) applied in medical setting (see Table 2). In fact, every patient needs to know his/her health status and to realize how physical training could improve it. It will be a physician’s responsibility/skill to drive the subject/patient from the general concept of “exercise is good for health” toward the full understanding of real individual benefits which he may derive from becoming physically active, following the exercise prescription tailored also on the assessment of results.

### 5.2. Goals Setting

It consists in the definition of the clinical goals that patients need to reach and considers both subjects’ initial requests (for instance to lose weight) and physician’s considerations derived from subjects’ assessment (for instance to loose fat mass and to improve muscular mass). Goal setting will determine the correct exercise prescription. For instance the optimal exercise dose defined from international guidelines is clearly aimed to promote and maintain health [18,20,28,48,96]. If the goal were different, for instance to manage back pain, the prescription should be different! In the specific example the exercise modality needs to consider flexibility exercise, to relax back muscles, and strength exercise to improve muscular mass and quality. Different clinical goals require different modality and/or intensity of exercise.

Figure 2 shows the exercise modalities needed to reach the main clinical goals. Addressing possible contraindications revealed by clinical assessment may become a further goal to reach. For instance the presence of lumbar bulging disks in an overweight subject may represent a contraindication to high impact exercise, such as running or brisk walking on treadmill, and suggests the need of specific exercise modality to manage/prevent lumbar pain. In this case swimming might be a best endurance choice and flexibility/strength exercise at low-moderate intensity involving lumbar/abdominal muscle would be also welcomed. To discuss with the subject how his/her assessments and clinical condition shape goals definition and subsequently tailor exercise prescription, is of paramount importance in order to foster his/her proactive role. The subject needs to realize the possible real benefits that he/she will obtain and the practical and psychological resources that are required.

### 5.3. Exercise Prescription

International Guidelines define the optimal dose of exercise to obtain health benefit [19,20,28,48,93]: in order to promote and maintain health, all healthy adults aged 18 to 65 years need moderate-intensity aerobic physical activity for a minimum of 30 min on five days each week or vigorous-intensity aerobic physical activity for a minimum of 20 min on three days each week. Combination of moderate- and vigorous-intensity activity can be performed to meet this recommendation [19,20,28,48,93]. In addition, every adult should perform activities that maintain or increase muscular strength and endurance for a minimum of two days each week [19,20,28,48,93,97]. Even low doses (a daily average of 15 min of moderate intensity) [23] of exercise may be associated to health benefit when comparing to sedentary individual. On the other hand, because of the dose-response relation between aerobic physical activity and health, to exceed the minimum recommended amount of physical activity will result in a further improvement of personal fitness, reduction in risk for chronic disease and disabilities [20,28,48,93,98]. Very high dose of exercise (generally characterized by anaerobic metabolism) were not associated to health benefits [98,99]. This concept is well pointed out in the most recent World Health Organization 2020 guidelines [28], which prefer to indicate a target range of 150–300 min of moderate-intensity and 75–150 min of vigorous-intensity physical activity (instead of simply achieve at least 150 min of moderate-intensity or 75 min of vigorous-intensity activity per week). This change recognizes that there is a range of physical activity which grants the maximal risk reductions for health outcomes associated with physical activity and going beyond this range does not further reduce the risk of major outcomes.

Recommendation for older subjects [19,28] are very similar to those for adults, although some differences particularly related to intensity levels and the set goals need to be considered. Moreover, multicomponent physical activity, which emphasizes functional balance and strength training to enhance functional capacity and prevent falls, are recommended [28].

Children and adolescent require a higher exercise dose corresponding to at least an average of 60 min/day of moderate to vigorous intensity physical activity [28], including muscle-strengthening and bone loading activities, performed at least 3 days a week [19,100]. Children before scholar age should spend at least 180 min in a variety of physical activities at any intensity including moderate-to vigorous- intensity physical activity (of which at least 60 min for children older than 3–4 years), spread throughout the day [101].

Tailored prescription of exercise programs is strongly recommended in all exercise guidelines [3,19,20,48,93,102] and requires the clear definition of modality, intensity, frequency, duration and progression of exercise. The concept of exercise dose or volume refers to the combination of intensity, duration and intensity.

#### 5.3.1. Exercise Modality: To Specify the Activity to Perform in Order to Reach the Set Goals

From a clinical point of view it may be important to define exercise modality in the context of physiological and biomechanical demands/types [93]:Endurance activity entails rhythmic motion of large muscle groups in aerobic activities (walking, jogging, swimming, etc.) [93]. Aerobic activity is physical exercise that depends primarily on the aerobic energy-generating process [60], it may be of low to high intensity depending on subject’s fitness level (i.e., exercise capacity). Endurance aerobic activities are primarily prescribed in order to improve cardiorespiratory fitness, to reduce cardio-metabolic-oncologic risk, to reduce fat mass, to improve wellness and to maintain health [3,19,20,48,93].Resistance or strength exercise involves activities that use low- or moderate-repetition movements against resistance [93], it is primarily prescribed in order to increases strength and muscle mass and physical independence. Conventional strength exercises typically consist of lifting heavier weights with longer rest periods (a greater anaerobic component), whereas circuit training consists of lifting lighter weights with shorter rest periods between exercises, introducing a greater aerobic component to the workout [97].Flexibility and muscle stretching exercises are focused on improving joint range of motion (flexibility), and on decreasing muscle tension [103], they are primarily prescribed in order to relieve muscle pain associated to muscle tension, to improve joint range and to prevent injuries.Balance exercise are aimed to improve the ability to maintain the body’s center of gravity within its base of support and it is primarily prescribed in order to reduce the risk of falls and injury [104].

#### 5.3.2. Exercise Intensity: To Define the Effort to Perform

Endurance aerobic exercise: (see Table 4) Ideally, to define the intensity of an aerobic endurance exercise, a cardiopulmonary exercise test (CPX) would be required in order to establish cardiorespiratory fitness (VO_2max_) and subsequently define the exercise intensity as percent of it. From a practical point of view, the exercise intensity is usually indicated by training heart rate, based on the approximate linear relationship between the increase of O_2_ consumption and the increase of heart rate (HR). Training heart rate may be calculated employing the heart rate reserve (HRR) formula (see Table 4) starting from resting heart rate and actual maximal heart rate measured by conventional maximal (not submaximal!) exercise stress test. If it is not possible to perform even this test, the usage of predefined tables to estimate the training HR might be useful, but this easier approach cannot be considered a tailored exercise intensity prescription (it is only a best guess!) and must not be used in case of patients under pharmacological chronotropic therapies (such as beta blockers) or in subjects particularly deconditioned [105]. Another solution, particularly when it is useless/impossible to perform a maximal exercise stress test (for instance in the initial phase of a training program of a sedentary/deconditioned subject who might be unable to reach maximal cardiovascular response, because of low exercise capacity) empirical methods may be considered, as general physical activity promotion tools (Table 4).

In order to get significant benefits from endurance aerobic exercise a moderate exercise intensity is generally required; even lower intensity may be useful particularly in the beginning of intervention programs or in deconditioned patients [23]. On the other hand, the benefits will increase performing exercise at higher intensities as long as they remain aerobic [35,106]. Increasing the exercise intensity means also to increase risk (particularly cardiac and musculoskeletal risk) [20,35] and at very high dose, endurance exercise will not be performed using a prevalent aerobic metabolic pathways [20,107]. Elevated intensity of exercise may be even associated to an increased mortality [99,106].

Strength exercise: to define the intensity of strength exercise is quite difficult. It depends from many factors, such as the resistance (generally represented by a weight to lift) that the muscle must win, the speed of movement and number of repetitions. Usually, there is an inverse association between the weight to lift and the speed of movement: the more elevated is the weight and the less is the speed of movement and/or then the number of repetitions that the subject may sustain. A lower repetition range with a heavier weight (anaerobic) may better optimize muscular strength and power, whereas a higher repetition range with a lighter weight may better enhance muscular endurance. Using weight loads that permit 8 to 15 repetitions will generally facilitate improvements in muscular strength and endurance [97]. To calculate strength intensity one-repetition maximum (1RM) (which is the maximum amount of weight that a person can possibly lift for one repetition, considered also as the maximum amount of force that can be generated in one maximal contraction) is generally employed [20,97]. 1RM can either be calculated directly using maximal testing or indirectly using submaximal estimation [108]. In order to better meet patients’ capabilities, the use of multiple repetitions, usually five (5RM) using a lighter weight, may be considered (5RM represents the maximum amount of weight that can be performed 5 times [20]). This technique helps to avoid maximal exercise which may be difficult to perform and possibly dangerous in risk populations.

#### 5.3.3. Exercise Duration: Defines How Long the Exercise Session Has to Last

This parameter is particularly important for endurance exercise and the session length required to improve health is of 30 min every day if the intensity is moderate. In unfit subjects who cannot exercise for 30 min consecutively, even shorter duration at moderate intensity may grant some benefits when compared with individuals who were inactive [23] particularly if the shorter period is repeated during the day. Recent guidelines [28] underline that physical activity of any bout duration is associated with improved health outcomes. Longer periods may be required to reach particular goals: for instance to loose fat mass 60 min/day of endurance moderate exercise is required [96].

#### 5.3.4. Exercise Frequency: To Define How Many Times the Subject Nneeds to Exercise

Endurance aerobic exercise needs to be performed ideally every day, at least 5 days/week, particularly if the intensity of exercise is light or moderate. Subjects who are fit and can exercise at aerobic vigorous intensity may also exercise 3 days/week [16,18,20,93,109]. On the other hand when the goal is to improve metabolic control in diabetic and obese patients a high frequency is welcomed [48,111].Strength exercise requires to exercise the same muscular group two-three, days/week, non-consecutive days in order to permit muscular recovery [48,97].

#### 5.3.5. Exercise Progression

To drive the subject toward the planned exercise dose in order to reach the established goal [48,93,102,112]. This is a particularly important point in order to foster subject’s compliance. Progression consists in modulating intensity, frequency and duration of exercise considering subject’s training level, preference and personal characteristic. For instance, for an obese, unfit patient 30 min of endurance aerobic moderate exercise performed every day may be a very difficult goal to reach. Starting with a less demanding protocol, and subsequently improving it, could be a better solution. The prescription of a tailored exercise program should be accompanied by a reduction of sedentary behavior (to benefit of any occasion during regular daily chores, or working time, to perform physical activity) [50,51,57]. Replacing sedentary time with any intensity of physical activity (including light intensity) may grant health benefits [28]. The reduction of sedentariness often represents the first step particularly in unfit subjects unwilling to perform structured exercise or unfit patients with an elevated cardiometabolic risk. In these subjects the prescription of a structured exercise program may represents a second step.

#### 5.3.6. Exercise Execution

From a medical and behavioral point of view, an exercise prescription may be considered correct only if the subject actually performs the prescribed exercise and this latter one is capable to improve physiological mechanisms such as endocrine, immunological and autonomic controls. Execution of exercise may occur in different places (home, fitness centers, outdoor, indoor, etc) considering subjects’ preferences/need. The presence of a health professional, such as exercise physiologist or physiotherapist, capable to transform into concrete actions the prescribed program, will tremendously improve the program and the results [65].

#### 5.3.7. First and Follow Up Visits

Table 5 depicts the main actions that may be performed during a first encounter or follow up visit with the subject/patient in order to prescribe an exercise program. Considering that exercise is only one component of a lifestyle intervention, to address also the others components, in particular nutrition, may be of pivotal importance in order to reach the defined clinical goal/s [61]. This action may be taken during the first encounter or in follow up visits, considering time constraint and subject’s/patient’s characteristics and preferences

## 6. Exercise to Manage Stress and Functional Syndromes

Exercise represents a preventive/therapeutical tool in many clinical conditions ranging from cardiometabolic disease to cancer. Being this Review part of a Special Issue entitled “Sport-Exercise and Stress: A Winning Combination” we consider meaningful to underline the role of exercise in the management of stress and conditions, such as functional syndrome, where stress may play an important role.

The definitions of stress are many, particularly considering the various interplay between psychological, physiological, behavioral or social aspects [113,114,115]. In a previous paper [25] we proposed the following definition in the full awareness that it might be improved “stress may be considered as the psychological, behavioral and physiological (or pathophysiological) consequence of the interaction between a subject and a stressor; considering as “stressor” everything (acute or chronic) present in the environment or in the subject’s mind that could be perceived as important, dangerous or potentially capable to modify, both negatively or positively, the subject’s life”. Stress per se [116] is a physiological response [114] to re-establish homeostasis through regulatory systems (hypothalamic-pituitary-adrenocortical, autonomic nervous system and immunity) [25,117,118,119,120] modulated by subjective perception of the stressor, individual (genetic, biological, psychological) differences and behavior [120]. The negative nature of stress manifests itself when bodily (somatic symptoms/diseases [115,119,121,122,123]) or psychosocial effects appear. These negative consequences sometimes are more determined by individual characteristics, as the personal way of perceiving the stressor and behavior [25,120] (coping strategy), then by the initial stressor. Sometimes, they may even worsen the clinical picture becoming themselves new stressors [25,124] (for instance the fear to be ill) and may even worsen the risk of occurrence of chronic non communicable diseases (assumption of unhealthy lifestyles). Moreover, subjects may, again unconsciously, change decision making strategies [25,125,126], change social behavior, isolating themselves, incrementing litigation in the family or on the job, etc, reducing their work and social performance thus further deteriorating their quality of life.

The mechanisms which link chronic stressful conditions to health are complex and modulated by genetic predisposition [25,114,119], being both direct (involving bodily control systems) and indirect (favoring unhealthy life styles). From a clinical point of view [25], chronic stressful conditions may play a role in the development/worsening of other diseases and in determining somatic symptoms not explained by the presence of “traditional” disease. These conditions are now defined as functional syndromes [123]. In fact, a somatic symptom may not always have an organic cause, but it might arise in consequence of an increased awareness of physiological changes associated with a stressor or disease/injury or physiological process, or variants of normal physiology. Individual experience and/or biological characteristics [25,123,124] may facilitate the occurrence of specific symptoms. Moreover the behavior assumed by the patient just to reduce the symptoms, may instead increase them in the long term, for instance to completely stop physical activity to reduce lower back pain [25].

From a clinical management point of view, stress and functional syndromes require a multidimensional approach which considers physiological, psychological and behavioral factors contributing to the genesis of symptoms [25,119,120,121,122,124,127,128].

A correct assessment in order to exclude other possible diseases and reach a clear diagnosis is, however, essential before starting treatment [129]; traditional tests and examinations are crucial to define the nature of reported symptoms and new diagnostic tools [25,129], as autonomic nervous system assessment, may furnish help in order to discover peculiar alterations present in patients who lament “unexplained” somatic symptoms and to monitor the effects of treatments [25,130,131,132]. Several studies show that Functional syndromes and stress conditions are actually characterized by an impaired cardiac autonomic control [130,131,132,133,134].

### 6.1. Management of Stress and Functional Syndromes 

The recent multidimensional approaches to manage stress [25,110,119,133,135] and other functional syndromes [25,124,136] consider the following tools:

#### 6.1.1. Explanation and Reassurance

A plausible explanation of symptoms, based on a correct diagnosis and available evidence, is offered by references [121,122] to provide a clear summary of the effects of predisposing (like preexisting diseases or genetic background), precipitating (like psychosocial stressors or infections [137,138] and perpetuating (like unhealthy lifestyle modifications or erroneous patient’s interpretation of personal condition) factors. The use of modern, simple diagnostic techniques capable of revealing autonomic nervous systems alterations may represent a useful tool [25] to explain that a symptom may be present even in absence of a traditional organic damage, being generated by a functional impairment [130,131]. This approach can be the starting point for translating a therapeutic rationale in concrete actions that also need patient’s active role such as lifestyle modification.

#### 6.1.2. Pharmacological and Psychological Treatments

Antidepressants and anxiolytics are the more frequently prescribed drugs even if their usefulness is controversial [124,139]. Other drugs aimed at peripheral physiological processes [128], for instance muscle tension, inflammation, bowel function [140] could be useful, in specific patients lamenting specific symptoms. On the other hand psychological treatments, in particular cognitive behavioral therapy and relaxation techniques [121,124,128,130,131,139,140,141,142,143,144] proved to be efficacious.

#### 6.1.3. Lifestyle Modifications

The assumption of an unhealthy behavior (improper coping strategy) may be a consequence of chronic stressful conditions (for instance lack of time for exercise) or an attempt to minimize symptoms and individual impairment in some functional syndromes (for instance to rest as much as possible in presence of muscular pain and or fatigue). A cornerstone of the treatment is to counterbalance these unhealthy behaviors. In this context, a particularly important role is played by physical activity [9,10,11,145,146]. Exercise may improve brain function, primarily cognition and executive control processes [146]. It may also reduce symptoms of anxiety and depression and it may improve stress system dysregulation [145]. Exercise, particularly aerobic activity [147], plays an important role in the management of unexplained somatic symptoms, chronic fatigue and fibromyalgia [119,121,124,136]; recent data [148] show also a role of strength exercise in the management of anxiety. The mechanisms underlying these beneficial effects are multifarious; of main importance may be the role of exercise in modulating all the control systems (immunological, hormonal and autonomic nervous system [4,37,38,39,41,43,44,46]) implied in the genesis of stress. Moreover the possibility to prove to the patient that he/she could modify his/her behavior (for instance to become physically active) may be a powerful tool to improve self-efficacy and self-esteem, mandatory skills in order to eventually take actions in order to manage the cause of stress and the stressful situation. Patient can realize that he/she can assume a proactive role, can regain the control on own life, can find the resources to take actions and change.

In order to manage stress and somatic symptoms, exercise must be correctly prescribed and tailored to patient’s characteristics and preferences [20,24,48,102]. Patients lamenting fatigue, muscular pain or other symptoms generally dislike performing physical activity and believe that it may be useless, if not even dangerous to their peculiar condition. Incorrect exercise prescription might strengthen these incorrect beliefs. A correct exercise protocol should include the establishment of a baseline followed by a planned increase in duration of low intensity physical activity, followed by gradual increases in intensity leading to moderate loads of aerobic exercise.

Modification in nutrition styles, smoke, sleep hygiene, are other lifestyle components which need to be taken into account, considering obviously patient’s characteristics and clinical condition.

## 7. Conclusions

The role of exercise to foster wellbeing, to support healthy ageing, to prevent and manage many chronic non communicable diseases and stress conditions, is nowadays undoubted. On the other hand to modify lifestyle introducing exercise into daily life is very difficult. To move from “what” to “how” [149], may be a keystone to realize exercise medicine. Physician’s competence to foster a patient’s behavioral change is important as much as technical skills to prescribe exercise. While this concept may seem self-evident, it is anything but established in current practice. To include lifestyle medicine (considering all the required competencies) in undergraduate medical curricula [67,68,69,71,150] or in other health professionals’ curricula, would be welcome in order to implement efficacious lifestyle change programs. The practical model proposed in this paper might represent a possible example to pioneer exercise medicine [62,63,64,65,66] as a component of everyday medical practice. Furthermore the importance that the physician could be a model (being or becoming a regular exerciser) for the patient needs to be underlined [112,151,152,153].

## Figures and Tables

**Figure 1 ijerph-18-00968-f001:**
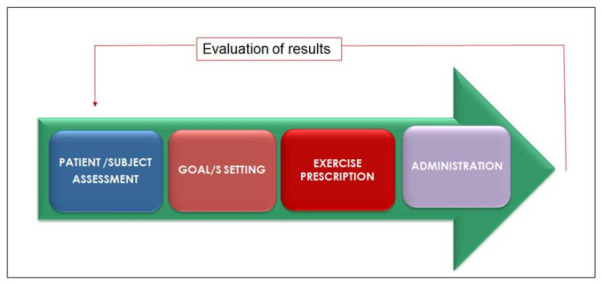
Process.

**Figure 2 ijerph-18-00968-f002:**
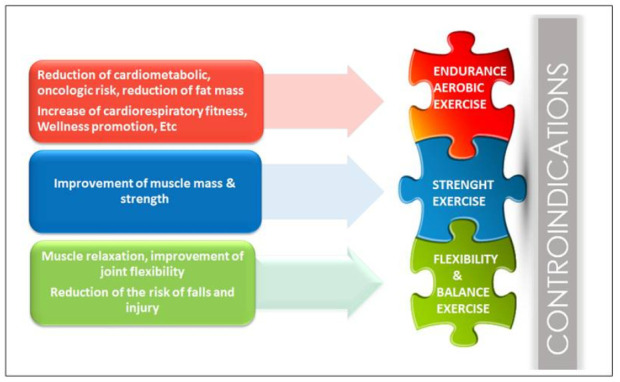
Exercise modalities needed to reach the main clinical goals.

**Table 1 ijerph-18-00968-t001:** Sport and exercise medicine: short glossary.

WELLNESS	The state of being in good health, especially as an actively pursued goal. It is an active process of becoming aware of and making choices toward a healthy and fulfilling life. It is more than being free from illness, it is a dynamic process of change and growth [52,53].
WELL-BEING	Positive outcome that is meaningful for people and for many sectors of society, it states that people perceive that their lives are going well [54].
HEALTH	State of complete physical, mental and social well-being and not merely the absence of disease or infirmity [55].
PREVENTION	Efforts aimed at avoiding a clinical event/disease [16]. - secondary prevention: efforts aimed at preventing the recurrence of clinical events in patients who have manifest clinical disease. - primary prevention: efforts focus on preventing the first occurrence of a clinical event among individuals who are at risk. - primordial prevention: to prevent the development of risk factors in the first place.
LIFESTYLE	Set of habits and customs that is influenced by the life-long process of socialization, including social use of substances such as alcohol and tobacco, dietary habits, and exercise, all of which have important implications for health [56].
SEDENTARINESS	Activities which typically are in the energy expenditure range of 1.0–1.5 METs, such as sitting during commuting, in the workplace, in the domestic environment, and during leisure time [57].
PHYSICAL ACTIVITY	Any bodily movement produced by skeletal muscles that results in energy expenditure [58].
EXERCISE	Subset of physical activity that is planned, structured, and repetitive and has as a final or an intermediate objective the improvement or maintenance of physical fitness [58].
PHYSICAL FITNESS	Set of attributes that are either health- or skill-related that people have or achieve. It is a multidimensional concept. The health-related components of physical fitness are: cardiorespiratory endurance, muscular endurance and strength, body composition and flexibility [58].
SPORT	An activity involving physical exertion and skill in which an individual or team competes against another or others for entertainment [59].
AEROBIC ACTIVITY	Ahysical exercise that depends primarily on the aerobic energy-generating process, it may be of low to high intensity depending on subject’s fitness level [60].

**Table 2 ijerph-18-00968-t002:** Cognitive behavioral strategies (Cbs) applied in medical settings [17,84,85].

**Educating the patient**	Informing the patient about the importance of lifestyle in determining cardiometabolic/oncologic risk, discussing personal risk and personal lifestyle assessment [86,87] as described by metrics derived from clinical examination, blood and instrumental tests, and ad hoc questionnaire.
**Determining an alliance with the patients and enhancing Self Efficacy**	Clarifying physician’s and patient’s role and responsibilities, and fostering patient’s proactive role in changing behavior giving special attention to personal strength.
**Realistic goal setting**	Determining priorities for behavior’s improvements based on medical requirements and personal preferences.
**Motivating patients**	Helping patients to find personal resources in order to transform generic and vague thinking into realistic, precisely defined goals so that the change in behavior could be the consequence of patient’s will and not of physician’s imposition.
**Self-monitoring**	Nutrition diary, sedentariness and physical activity monitoring before and during treatment.
**Problem solving**	Helping patients to identify possible barriers to improve lifestyle, brainstorming solutions with their pros and cons, and drawing a realistic action plan.
**Feedback and reinforcement**	Discussing positive and negative results at subsequent follow ups, helping patients to discover new strategies and resources and furthering transformation with deep education in order to continuously improve lifestyle.

**Table 3 ijerph-18-00968-t003:** Main psychological theories and models of behaviour change.

**Self-Efficacy**	Considers that an individual’s belief that he/she has the capabilities to produce an effect or reach a certain goal is a major determinant of behavioral change.
**Transtheoretical Model** [89] (also referred to as the ‘Stages of Change’ model)	Suggests to tailor the intervention to individual stage of change: precontemplation (the subject is unaware of the problem and he/she is not even considering changing), contemplation (the subject is aware of the problem and he/she is ambivalent about changing), preparation (the subject intends to take action and is prepared to experiment with small changes), action (the subject takes definitive action to change), maintenance and relapse prevention (the subject works to sustain the behavior change over the long term).
**Social Cognitive Theory** [90]	Focuses on the role of observing and learning from others, and on positive and negative reinforcement of behavior.
**Theory of Planned Behavior** [91]	Assumes that people’s behavior is determined by intention, and is predicted by attitudes, subjective norms (beliefs about whether other people approve or disapprove), and perceived behavioral control (beliefs about whether it is easy or difficult to do).
**Self Determination Theory** [92]	Combining skill development with underlying, intrinsic motivation and reason, is believed essential for lasting change. Intrinsic motivation does not rely on external pressure (for instance rewards or punishment from other people, but it exists within the individual, and is driven by interest or enjoyment in the task itself. People need to feel a sense of choice and responsibility for their actions, to feel capable of achieving the goal and also understood, cared for, and valued by others.

**Table 4 ijerph-18-00968-t004:** Classification of physical activity intensity.

		Intensity					
	Effort	Light	Light	Moderate	Hard	Very Hard	Maximal
**RELATIVE INTENSITY**	VO_2_max (%) HRR * (%)	<25	25–44	45–59	60–84	≥85	100
	Maximal HR (%)	<30	30–49	50–69	70–89	≥90	100
	RPE	<9	9–10	11–12	13–16	>16	20
**ABSOLUTE INTENSITY**	INTENSITY	Sedentary **	Light	Moderate	Vigorous		
	METs	1–1.5	1.6–2.9	3.0–5.9	6–9	≥9	
**Empirical tools**		▪ at rest with limited added movement	▪ no noticeable change in breathing ▪ can sustain activity for 1 h or more	▪ breathing is faster but compatible with speaking full sentences, but not singing ▪ increased sweating ▪ can sustain activity for 30–60 min ▪ walking at least 100 step·min–1	▪ feeling “out of breath”, breathing very hard, incompatible with carrying on a conversation comfortably ▪ increased sweating ▪ can sustain activity for up 30 min	▪ feels like giving 100% ▪ All out burst of between 1 and 2 min ▪ intensity cannot be sustained for more than 10 min	▪ it is impossible to sustain the effort
**Examples**		▪ sittings ▪ reclining or lying ▪ watching TV ▪ using a PC ▪ driving a car	▪ walking slowly ▪ light work while standing ▪ playing an instrument	▪ brisk walking (5–6 Km/h) ▪ slow cycling (15 Km/h) ▪ ballroom dancing ▪ recreational swimming	▪ jogging ▪ running ▪ biking > 15 km/h ▪ swimming laps ▪ single tennis	▪ straining or competing in most competitive sports ▪ racing or any all-out activity (eg running, rowing, swimming, skiing and high intensity intervals)	
**Training zone *****		Aerobic	Aerobic	Aerobic	Aerobic + Lactate	Aerobic + Lactate + Anaerobic	Anaerobic

Modified from: [18,20,62,93,109,110]. VO_2_max = maximal aerobic capacity; HRR = Heart rate reserve = Maximal heart rate —resting HR; METs indicates metabolic equivalents. 1 MET = 3.5 mL O2 kg^−1^ min^−1^. * % Heart rate reserve (HRR) = calculate HRR target by (HRR × %value) + resting HR; RPE, rating of perceived exertion (20 value Borg score); ** Sedentary behavior is defined as any walking behavior which is characterized by an energy expenditure ≤ 1.5 Mets. *** Adapted from refs [105], using training zones related to aerobic and anaerobic thresholds. Low-intensity exercise is below the aerobic threshold; moderate is above the aerobic threshold but not reaching the anaerobic zone; high intensity is close to the anaerobic zone; and very intense exercise is above the anaerobic threshold. The duration of exercise will also largely influence this division in intensity [20].

**Table 5 ijerph-18-00968-t005:** Description of first and follow up visits (see also our previous paper [4]).

1ST VISIT ACTIONS	Itemized AIMS
**Welcome**	▪ Understand patient’s concerns and circumstances ▪ Introduce the program and define patient’s and physician’s roles & responsibilities ▪ Establish an empathic, maieutic relationship
**Clinical history and clinical assessment**	▪ Definition of patient’s health status ▪ Collection of anthropometric data ▪ Take note of pharmacological plan and discuss about potential side effects ▪ Discussion of life style assessment ▪ If clinically relevant, prescription of further clinical tests
**Explanation of diagnosis and of specific benefits derived from lifestyle change**	▪ Make the patient aware about his/her clinical condition ▪ Support the patient in learning how lifestyle change may ameliorate his/her health and reduce cardio/metabolic/oncologic risk, using own history ▪ Allow the patient to elicit desire, motivation to change ▪ Give the opportunity to the patient to raise any question in order to have tailored answers
**Setting of specific individual goals**	▪ Allow the patient to express his/her own reasons for change ▪ Ensure the patient that she will have all the required support ▪ Prioritize behavior/issues for change (patient-physician alliance) ▪ Define long term goal/s and steps (short term goals) to reach it/them
**Education about physical activity and tailored exercise prescription**	▪ Provide knowledge and skills for exercise ▪ Be sure that the patient understands the importance of reducing sedentary behaviors and of increasing structured exercise ▪ Help the patient to discover and define practical strategies to reduce sedentariness during his/her normal daily activities ▪ Ensure that the patient realizes the importance of exercise and to appreciate any single improvement” ▪ Elicit in the patient the desire to adhere to the tailored exercise prescription taking into account medical guidelines, personal clinical needs and patient’s preferences ▪ Explain the importance of other components of lifestyle (nutrition, exercise, sleep, stress) and their relationship with exercise habits.
**FOLLOW UP VISITS** **ACTIONS**	**Itemized AIMS**
**Welcome**	▪ Reinforce the empathic, maieutic relationship
**Clinical history (from previous encounter to present) and clinical assessment**	▪ Assess changes in clinical status and life style from previous visit to present ▪ Collection of present anthropometric data ▪ Discussion of tests results prescribed during, or after first visit
**Analysis of results and of encountered barriers**	▪ Provide feedback on progression towards set goals ▪ Empower the patient and reflect on his/her central role in determining own health status and the success of the life style change program ▪ Reinforce the collaborative relationship with the physician and the possibility to get all the required support ▪ Make the patient aware of the (even small) result/s obtained and how it/them has/have improved his/her quality of life
**Problem solving**	▪ Help the patient to find solutions to overcome the encountered problems ▪ Help the patient to discover the required resources and to select those which best account for his/her needs ▪ Raise patient awareness of consequence of his/her behavior on his/her quality of life and wellbeing ▪ Make the patient aware that to encounter barriers and to find the resources to overcome them is a critical, always present, part of a life style change program
**Setting of further specific individual goals**	▪ Assist the patient to define further steps towards the long term goal ▪ Make the patient aware that he/she is regaining control on his/her life and wellbeings

## Data Availability

Not applicable.

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
