# Peer review of "Exercise Prescription to Foster Health and Well-Being: A Behavioral Approach to Transform Barriers into Opportunities"

_ijerph, 2021, doi:10.3390/ijerph18030968_

Round 1

Reviewer 1 Report

This is a well written review addressing the barriers of exercise prescription and intervention and potential solutions in practice. I just have several minor suggestions: 1) there are too many points in this review. It would be better to focus on the part 5 and shorten the part 6. It does not make sense to just put an emphasis on the management of stress and functional syndrome, but not on other conditions like type 2 diabetes, obesity and cancers.  In addition, the authors already have published a related paper (ref 25). 2) I did not find the citation of the Scheme II in the main text. 3) There are two typos in Scheme I. Please correct the "WELNESS" as "WELLNESS", the "WEEL-BEING" as "WELL-BEING".

Author Response

Answer to Reviewer 1:

We thank this Reviewer for his/her suggestions.

  1. We agree with the Reviewer 1 who wonders about the reason to put emphasis on the management of stress and functional syndromes and not on other very important clinical conditions. We highlighted Stress and functional syndromes because this Review will be part of a special Issue entitled "Sport-Exercise and Stress: A Winning Combination" https://www.mdpi.com/journal/ijerph/special_issues/sport_exercise_stress.In order to clarify this point we add the following sentences at the beginning of point 6….. “Exercise represents a preventive/therapeutical tool in many clinical conditions ranging from cardiometabolic disease to cancer. Being this Review part of a Special Issue entitled "Sport-Exercise and Stress: A Winning Combination" we consider meaningful to underline the role of exercise in the management of stress and conditions, such as functional syndrome, where stress may play an important role.” Lines 548-552
  2. We have also shorted point 6 summarizing previous section 6.1.2 e 6.1.3 in only one paragraph which now is 6.1.2
  3. Authors considering meaningful, from one hand to evidence the importance of behavioral aspects in the prescription of exercise, and from another hand, to underline the role of exercise in the management of condition characterized by stress.
  4. We thank the Reviewer 1 to have notice this point. We (see Scheme II), line 240
  5. We thank the Reviewer 1 to have notice these typos. We corrected as suggested, lines 103 and 107
  6.  

Reviewer 2 Report

Line 13: amend to “but considered by some unrealistic in current”

Line 17, Consider rewording: These range from traditional clinical competencies, to technical competencies to correctly, prescribe exercise, to competencies in behavioral medicine to motivate the subject.

Please review the whole manuscript for incorrect spacing

Line 153: “also other component(s) of lifestyle”

Line 221: “present(ed) in clinical contexts”

Line 222 “patients regard(ing) the financial burden”

Line 229 “more th(a)n on”

Line 361: “If the goal would be different, for instance to improve body composition loosing fat mass and increasing lean mass, the prescription would be different! In the specific example the exercise dose of endurance exercise would be doubled![96].” I would consider using an different example as it is not entirely accurate as you can loose fat mass and increase lean mass using other exercise modalities such as resistance exercise.

Line 409: “easily performed using also anaerobic metabolic pathways” Please simplify the description of anaerobic metabolic pathways in this example in this for readers.

Line 435 “Aerobic (a)ctivity”

463 Please give some examples of pharmacological chronotropic therapies for the reader ie. beta blockers.

Line 479 “that the muscle must win, the speed of movement and number of repetitions. Usually, the greater the resistance to win the less is the speed of movement and then the number of repetitions, resulting in a higher relative effort” Please rephrase as ambiguous

Line 489 “meet patients’ capabilities” Please expand on the evidence on the use of the 5RM instead of the 1RM in terms of safely testing in high risk populations

Line 530 Exercise (e)xecution

Table 2 Clinical history and clinical assessment section  “if the case” consider rewording to if clinically relevant.

Table 2 Explanation of diagnosis and of specific benefits derived from lifestyle “Change clear up facts from patient’s interpretation of the personal implications” consider deleting this statement as it overlaps all other statements in this section

 Table 2Education about physical activity and tailored exercise prescription  “may be capable of appreciating any single improvement” Consider rephrasing to “and to appreciate any single improvement”

Author Response

Answer to Reviewer 2:

We thank this Reviewer for his/her suggestions.

  1. We changed line 13 as per her/his suggestion
  2. We reword the following sentence: “Physicians need specific skills to improve patients’ exercise habits. These range from traditional clinical competencies, to technical competencies to correctly, prescribe exercise, to competencies in behavioral medicine to motivate the subject.” In “Physicians need specific skills to improve patients’ exercise habits: traditional clinical competencies, technical competencies to correctly prescribe exercise, and competencies in behavioral medicine to motivate the subject.”  Lines 16-18
  3. We thank the Reviewer 2 for having noticed incorrect spacing, that we we have correct
  4. Line 153: “also other component(s) of lifestyle”: we have corrected. Line 152
  5. Line 221: “present(ed) in clinical contexts”: we have corrected. Line 220
  6. Line 222 “patients regard(ing) the financial burden : we have corrected Line 221
  7. Line 2(9)9 “more th(a)n on”. we have corrected Line 300
  8. Line 361: “If the goal would be different, for instance to improve body composition loosing fat mass and increasing lean mass, the prescription would be different! In the specific example the exercise dose of endurance exercise would be doubled![96].” I would consider using an different example as it is not entirely accurate as you can loose fat mass and increase lean mass using other exercise modalities such as resistance exercise.
  9. We substitute the example with another one: “If the goal would be different, for instance to manage back pain, the prescription would be different! In the specific example the exercise modality needs to consider flexibility exercise, to relax back muscles, and strength exercise to improve muscular mass and quality.” Lines 361-364
  10. Line 409: “easily performed using also anaerobic metabolic pathways” Please simplify the description of anaerobic metabolic pathways in this example in this for readers.
  11. We change the sentence with the following one: …very high dose of exercise (generally characterized by anaerobic metabolism) were…. Line 410
  12. Line 435 “Aerobic (a)ctivity”. We have corrected   line 435
  13. 463 Please give some examples of pharmacological chronotropic therapies for the reader ie. beta blockers.
  14. We modified the sentence as follows: “….under pharmacological chronotropic therapies (such as beta blockers) or….” Line 463
  15. Line 479 “that the muscle must win, the speed of movement and number of repetitions. Usually, the greater the resistance to win the less is the speed of movement and then the number of repetitions, resulting in a higher relative effort” Please rephrase as ambiguous.
  16. We modified the sentence as follows: “It depends from many factors, such as the resistance (generally represented by a weight to lift) that the muscle must win, the speed of movement and number of repetitions. Usually, there is an inverse association between the weight to lift and the speed of movement: the more elevated is the weight and the less is the speed of movement and/or then the number of repetitions that the subject may sustain”. Lines 477-481
  17. Line 489 “meet patients’ capabilities” Please expand on the evidence on the use of the 5RM instead of the 1RM in terms of safely testing in high risk populations
  18. We modified the sentence as follows: In order to better meet patients’ capabilities, the use of multiple repetitions, usually five (5RM) using a lighter weight, may be considered (5RM represents the maximum amount of weight that can be performed 5 times [20]). This technique helps to avoid maximal exercise which may be difficult to perform and possibly dangerous in risk populations. Lines 489-493
  19. Line 530 Exercise (e)xecution. We have corrected. Line 533
  20. Table 2 Clinical history and clinical assessment section “if the case” consider rewording to if clinically relevant.
  21. We have modified the table to as per suggestion
  22. Table 2 Explanation of diagnosis and of specific benefits derived from lifestyle “Change clear up facts from patient’s interpretation of the personal implications” consider deleting this statement as it overlaps all other statements in this section
  23. We modified the section deleting the sentence as reviewer’s suggestion
  24. Table 2Education about physical activity and tailored exercise prescription “may be capable of appreciating any single improvement” Consider rephrasing to “and to appreciate any single improvement”
  25. We modified the section modifying the sentence as reviewer’s suggestion
  26.  
